# Guideline-based bismuth quadruple therapy for helicobacter pylori infection in China: A systematic review and network meta-analysis

Jiali Wei[1¤a☯], Zehao Zheng[1☯], Xin Wang[2], Boyi Jia[3], Mingyao Sun[1], Jiayi Wang[1], Qin Wan[1], Mei Han[4¤b*], Yue Qiu[5*]

**1** Beijing University of Chinese Medicine, Beijing, China, **2** Wang Jing Hospital of CACMS, Beijing, China, **3** Fangshan Hospital Beijing University of Chinese Medicine, Beijing, China, **4** Center for Evidence-Based Medicine, Beijing University of Chinese Medicine, Beijing, China, **5** The Third Affiliated Hospital of Beijing University of Chinese Medicine, Beijing, China

☯ These authors contributed equally to this work.
¤a Current Address: Beijing University of Chinese Medicine, Beijing, China.
¤b Current Address: Centre for Evidence-Based Medicine, Beijing University of Chinese Medicine, Beijing, China.
* hanmeizoujin@163.com (MH); springqiu@126.com (YQ)

## Abstract

### Background

Currently, quadruple therapy is unanimously recommended as the current first-line treatment option for Helicobacter pylori (H. Pylori) eradication. However, the drug composition of quadruple therapy is very complex, and the efficacy and safety between them is not clear.

### Aims

To compare the efficacy and safety of H. Pylori eradication regimens, which were recommended by the Fifth Consensus of China, by network meta-analysis.

### Methods

Literature databases were used to retrieve clinical randomized controlled trials of H. Pylori eradication. Network meta-analysis was performed using BUGSnet and meta package of R software, using OR values as effect size, and SUCRA was used to rank the efficacy of interventions under each outcome.

### Results

A total of 55 studies and 130 arms were included. The NMA analysis found that the top regimen in term of eradication rate outcome was: Rabeprazole + Bismuth + Furazolidone + Tetracycline (SUCRA, 97.5). In terms of safety outcomes: Lansoprazole + Bismuth + Amoxycillin + Clarithromycin (SUCRA, 91.97).

**Data availability statement:** All relevant data are within the article and its supporting information files.

**Funding:** This study was supported by High-level traditional Chinese medicine key subjects construction project of National Administration of Traditional Chinese Medicine——Evidence-based Traditional Chinese Medicine (zyyzdxk-2023249) and the Basic Research funding of Beijing University of Chinese Medicine. The funders had no role in study design, data collection and analysis, decision to publish, or preparation of the manuscript.

**Competing interests:** The authors declare that there are no conflicts of interest.

**Abbreviations:** H. pylori, Helicobacter pylori; PPI, Proton-pump inhibitor; B Bismuth; SUCRA Surface under the cumulative ranking; RUT, Rapid Urease Test; CMT, Culture and Molecular Testing; NR, No Information; AML, Amoxycillin; CLR, Clarithromycin; FRZ, Furazolidone; LEV, Levofloxacin; MTZ, Metronidazole; TE, Tetracycline; E1, Esomeprazole+B+AML+CLR; E2, Esomeprazole+B+AML+FRZ; E3, Esomeprazole+B+AML+LEV; L1, Lansoprazole+B+AML+CLR; L3, Lansoprazole+B+AML+LEV; O1, Omeprazole+B+AML+CLR; O3, Omeprazole+B+AML+LEV; P1, Pantoprazole+B+AML+CLR; P3, Pantoprazole+B+AML+LEV; R1, Rabeprazole+B+AML+CLR; R2, Rabeprazole+B+AML+FRZ; R3 Rabeprazole+B+AML+LEV; R8, Rabeprazole+B+FRZ+TE.

## Conclusions

The bismuth quadruple therapies recommended by the guidelines for the treatment of H. Pylori have good performance in terms of four-week eradication rate and safety outcome indicators, and due to the different resistance of antibiotics in different regional populations, rational use of drugs should be combined with local conditions.

## Introduction

Helicobacter pylori (H. pylori) is a highly mobile spiral Gram-negative bacterium found on the surface of the epithelial lumen of the stomach. The prevalence of H. pylori infection is about 50% globally. Besides, because of geographical differences, social economic, sanitary conditions and human differences, H. pylori infection rates are as high as 80% in poor and developing countries and 20-50% in developed countries [1,2]. A 2010 epidemiological study in China found that the prevalence of H. pylori infection was about 40-90%, with an average of 59% [3]. H. pylori infection is associated with a variety of diseases, and The Fifth Consensus Report in China [4] states that almost all H. pylori-infected patients have chronic active gastritis and most patients are asymptomatic. According to the World Health Organization (WHO), H. pylori infection causes about 75% of gastric cancers and 5.5% of other cancers, and the WHO has included H. pylori in its list of 16 antibiotic-resistant bacteria that pose the greatest threat to human health [5,6]. Because of this, rapid diagnosis and treatment of H. pylori infection in its early stages play an important role in curbing the spread of H. pylori infection. H. pylori can be diagnosed using non-invasive methods like the urea breath test, stool antigen test, and serology, or invasive methods such as endoscopy, histology, rapid urease test, culture, and molecular testing [7–9].

The treatment of H. pylori infection has become increasingly complex due to antibiotic resistance, and the eradication rate of H. pylori with standard triple therapy (PPI + two antibiotics) is decreasing as patients become more resistant to antibiotics. Because of this, the Kyoto Consensus [10], the Toronto Consensus Opinion [11], the Maastricht-5 Consensus [5] and the 2011 ACG [12] unanimously recommended quadruple therapy as the current first-line treatment option for the eradication of H. pylori, mainly divided into quadruple therapy with and without bismuth. The Fifth Consensus Report in China [4] also recommended seven bismuth quadruple regimens as first-line treatment, such as PPI + bismuth + (tetracycline + metronidazole/furazolidone, amoxicillin + levofloxacin/clarithromycin/furazolidone/metronidazole/tetracycline). In consideration of most current guidelines, we recommend quadruple therapy as an eradication of H. Pylori's first-line regimen. This view aims to incorporate clinical randomized controlled studies of H. Pylori eradication regimens recommended by the Fifth Consensus of China [4] by means of network meta-analysis to compare their efficacy and safety. Furthermore, the Fifth consensus report in China [4] indicates that resistance rates in China are 20–50% for clarithromycin, 40–70% for metronidazole, and 20-50% for levofloxacin. Due to varying levels of antibiotic resistance and environmental factors, regional differences can significantly impact the effectiveness of H. pylori eradication therapies. For instance, the prevalence and acquisition patterns of H. pylori infection often correlate with socioeconomic and demographic factors, such as population density, hygiene standards, and racial composition [13]. These differences suggest that treatment outcomes may not be uniform across regions. Due to the geographical diversity in China, which encompasses variations in healthcare access, dietary habits, and antibiotic usage, this study will conduct a subgroup analysis based on different geographic regions within the country.

## Materials and methods

This review was registered in the PROSPERO database (registration ID: CRD42022331163).

### Search strategy

A professional search was conducted in PubMed, the Cochrane Library, Web of Science, SinoMed, China National Knowledge Infrastructure Database (CNKI), China Science and Technology Journal Database (VIP) and China Wanfang Database. The language of the literature was limited to Chinese and English. The search deadline was November 24, 2023. The detailed literature search strategy was shown in S1 Table.

### Eligibility criteria

RCTs that met the following criteria were included.

(a) Type of study: Randomized controlled trials (RCTs).

(b) General characteristics of the patients: Mainland Chinese adults with confirmed H. pylori (+) diagnosis, regardless disease duration and gender.

Diagnostic methods: C13 breath test, C14 breath test, Rapid Urease Test and Molecular Testing and Endoscopy, etc.

(c) Interventions: Bismuth quadruple treatments for H. pylori infection are recommended by The Fifth Consensus Report in China (2017) [4].

(d) Comparison: Bismuth quadruple treatments for H. pylori infection are recommended by The Fifth Consensus Report in China (2017) [4].

(e) Outcomes: Primary outcome indicators: H. pylori eradication rate (four weeks after treatment). Safety outcome indicators: Incidence of adverse events (description of specific adverse events).

### Exclusion criteria

Studies that met any of the following criteria were excluded.

(a) Duplicate publications.

(b) Unavailability of the full text.

(c) Absence of data on the H. pylori eradication rate.

(d) Unclear timing of the measurement of the H. pylori eradication rate.

### Study selection

Two researchers conducted an initial screening of topics and abstracts according to the inclusion criteria. After checking, they read the full text for screening separately. If they had different opinions during the screening process, a third researcher would participate in the decision of inclusion.

### Data extraction

Two researchers individually extracted the following information. They extracted the characteristics of the patient (age, gender, whether first treatment or not, diagnostic methods). In addition, they extracted information about the methodology methods of random sequence

generation, allocation concealment methods, blinding - investigators, blinding -participants, blinding – data collectors and analysts, numbers and reasons for withdrawing and loss to follow up). They also extracted detail regarding the intervention (extraction of PPI and antibiotic combinations that were recommended in The Fifth Consensus Report in China [4]). Furthermore, they extracted information regarding outcome indicators (Four-week eradication rate, incidence of adverse events) and a description of adverse events (a description of the adverse events that occurred after the patient was administered the drug, such as nausea, vomiting, abdominal pain and so on). There were also some additional remarks (location of China, funding of the study).

## Risk of bias assessment

The Cochrane collaboration's tool for assessing the risk of bias in randomized trials (RoB2) will be used to assess the quality of the included studies. Two researchers need to measure five dimensions: the process of randomization, the process of assigning the intervention, deviations from the established intervention and their impact, missing outcome data, outcome measures, and selective reporting of outcomes. They will make judgements on a number of different signal questions under each domain and answer them objectively. The signal questions generally have five optional answers: Yes (Y), Probably Yes (PY), Probably No (PN), No (N), and No Information (NI), and the evaluation tool gives a risk outcome based on the responses. The two researchers will check the quality rating of the studies to be included as low risk of bias, some concerns, or high risk of bias. The risk of bias diagram and the risk of bias summary diagram are used to represent the risk of bias [14–16].

## Analysis methods

A Bayesian model approach was used for the network meta-analysis, which is done using Markov-chain Monte-Carlo (MCMC) [17]. The outcome indicators used in this study were eradication rate and adverse event rate, both of which were dichotomous variables. We used Odds Risk (OR) as the effect measure [18,19]. The selection of random and fixed effects models is based on the deviance information criterion (DIC). The DIC values of the random-effects model and fixed- effects model were calculated separately, and the model with the smaller DIC value was selected [17,18]. We used SUCRA (surface under cumulative ranking) to rank the interventions for each outcome. All statistical heterogeneity was derived from clinical and methodological heterogeneity. We quantified heterogeneity primarily using statistical heterogeneity: $I^2$, which was conducted by using the "gemtc" package in R software. We consider $I^2 < 50\%$ as low heterogeneity, $I^2 > 50\%$ as high heterogeneity [20]. We will conduct subgroup analysis for different regions in mainland China as these regions have different resistance to antibiotics [17,21,22]. Network Meta Analyses are carried out in R software using the BUGSnet and meta package [23]. Due to the large number of interventions in multiple combinations of quadruple therapy, we included interventions with more than five occurrences in the network meta-analysis study. A two-tailed $p < 0.05$ was considered statistically significant.

## Results

### Study selection

A total of 34,007 studies were retrieved in PubMed, the Cochrane Library, Web of Science, SinoMed, CNKI, Wanfang, and VIP, and 155 studies were obtained for full-text screening after

initial screening using NoteExpress software. A total of 91 studies were obtained after full-text screening, and after full-text data extraction, we included more than 13 interventions (more than five times) in this network meta-analysis with a total of 55 articles and 130 arms. The characteristics of all inclusion and exclusion literatures can be found in S2 Table. The flow diagram of study retrieval and selection is shown in Fig 1.

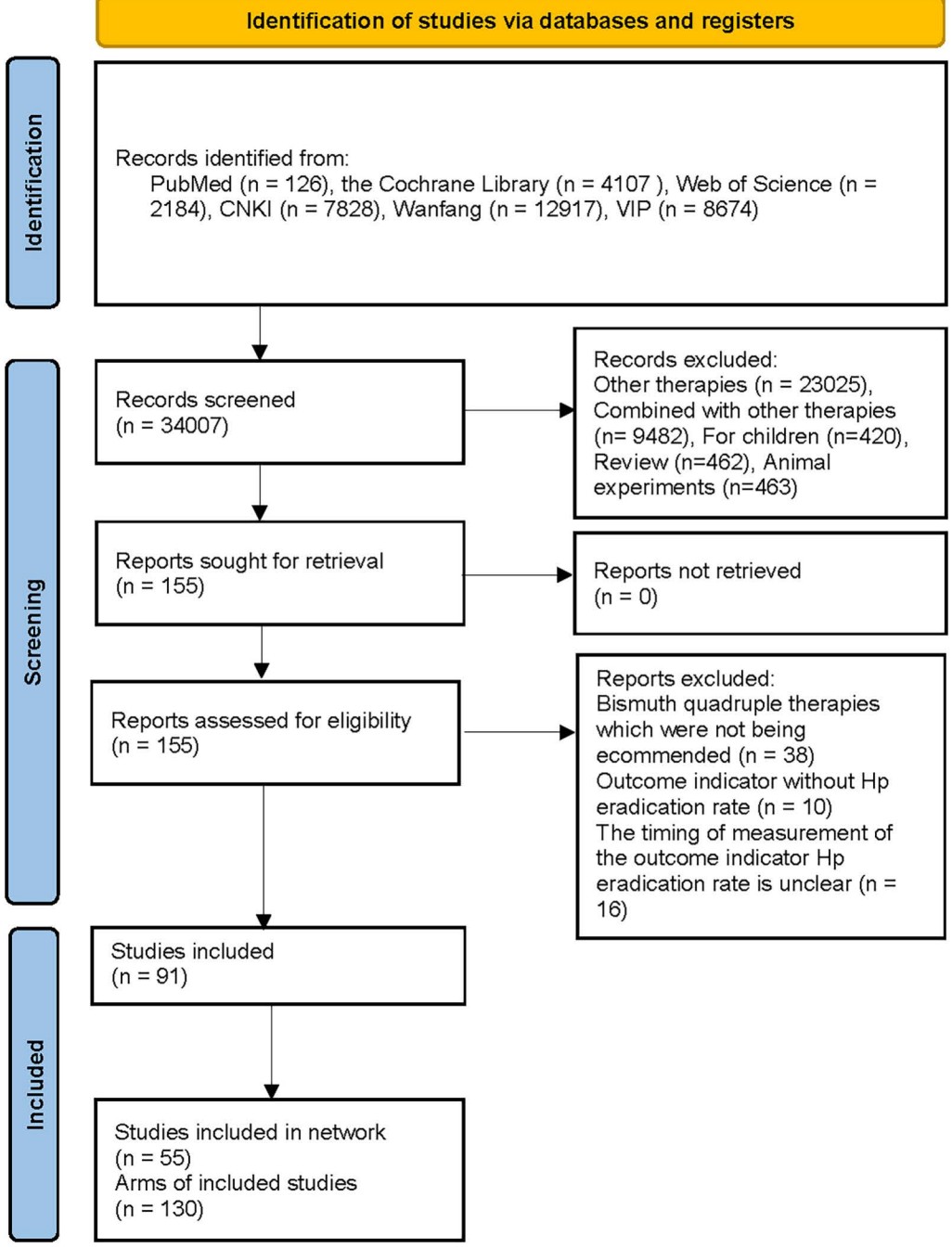

**Fig 1. Flowchart of study search and screening.** This diagram outlines the steps involved in identifying, selecting, and excluding studies for inclusion in the analysis, including the number of studies at each stage of the process.

## Study characteristics

In the included studies, the baseline of age and male-to-female ratio were consistent, and a total of 11,682 patients were included in the network analysis. Most studies did not explicitly mention whether the patient got treatment for the first time or if an antibiotic sensitivity test was performed. The main diagnostic modalities in the included studies were C13, C14, Rapid Urease Test (RUT), Culture and Molecular Testing, and Endoscopy. A total of 13 types of interventions for PPI + B + two kinds of antibiotics were included. As for the dosage and duration of treatment, according to the dosage recommended in the guidelines, the duration of treatment is usually 10-14 days because of the restrictions on the use of antibiotics [4]. The characteristics of included randomized controlled trials can be found in Table 1.

## Risk of bias assessment

**Domain 1: Randomization process.** The randomization method was mentioned in all included studies, with some studies explicitly stating the use of random number tables, and a few studies describing the use of opaque sealed envelopes for stochastic protocol concealment. The included studies were consistent across group baselines. Only Kong SY 2020 had a randomized scheme of concealment, so Kong SY 2020 was rated as Low Risk in Domain 1, and the remaining studies were all Some Concerns.

**Domain 2: Deviations from intended interventions.** Since Helicobacter pylori eradication therapy requires the use of multiple drugs, both patients and doctors can determine the treatment regimen used by patients, and none of the included studies were double-blind studies. None of the included studies mentioned patients switching groups as a result. Since the data in the included studies are inconsistent, the included studies will be rated as Low Risk in Domain 2.

**Domain 3: Missing outcome data.** The included studies had consistent data and described no drop-off and loss of follow-up. Therefore, the included studies were rated as Low Risk.

**Domain 4: Measurement of the outcome.** The diagnostic methods of pylori eradication in the included studies were C13 and C14, and the measurement methods were objective, so each study was evaluated as Low Risk in Domain 4.

**Domain 5: Selection of the reported results.** None of the included studies had published protocols. However, the eradication rates were all measured four weeks after the end of treatment in studies. Therefore, we considered that the included studies were less likely to report selective outcomes, so the included studies were assessed as Low Risk in Domain 5.

**Domain 6: Overall.** Finally, we rated Kong SY 2020 as Low Risk, others were Some Concerns overall.

The results of the risk of bias assessment for all included trials are summarized in Fig 2. The results of the risk of bias assessment for each domain of the included trials can be found in S3 Table.

## Network meta-analyses of primary outcome

A total of 13 interventions were included, as shown in Fig 3 (A). The SUCRA of the included interventions were: R8 97.58, R2 85.31, E2 78.11, R3 74.2, E3 66.1, L3 49.94, E1 47.89, R1 45.99, L1 30.7, P1 22.2, O3 21.96, P3 20.9, O1 9.1. The ranking of interventions was shown in Fig 4 (A). The league table of network meta-analysis in Fig 5.

## Network meta-analyses of safety outcome

A total of 13 interventions were included, as shown in Fig 3 (B). In terms of safety, the SUCRA of the included interventions were: L1 92.48, L3 84.12, E3 76.57, R1 67.75, R8 50.76, R3 49.96, E2 46.96, E1 41.15, R2 37.14, P1 37.14, O1 34.93, O3 23.14, P3 7.88. The ranking of

**Table1. The characteristics of included RCTs.**

| Study ID | Age (Group 1) | Age (Group 2) | Female/Male (Group 1) | Female/Male (Group 2) | Method of Diagnosis | Intervention (Group 1) | Intervention (Group 2) | Duration (Group 1) | Duration (Group 2) |
|---|---|---|---|---|---|---|---|---|---|
| Bai C 2019 [24] | 52.98±10.63 | 55.00±9.71 | 23/26 | 14/19 | C13 | R1 | R8 | 14d | 14d |
| Chen JE 2018 [25] | 45.6±10.5 | 48.9±9.7 | 24/27 | 28/23 | C13 | O1 | O3 | 14d | 14d |
| Chen JE 2018 [25] | 44.8±10.3 | | 27/24 | | C13 | O5 | | 14d | |
| Chen JX 2018 [26] | 43.6± 9.5 | 44.3±10.8 | 32/28 | 36/24 | C14 | O1 | L1 | 14d | 14d |
| Chen W 2013 [27] | 41.74±10.50 | 40.50±10.93 | 33/20 | 30/18 | RUT or C13 | E1 | E2 | 7d | 7d |
| Chen W 2013 [27] | 41.45±10.61 | | 32/19 | | RUT or C13 | E2 | | 7d | |
| Chen XY 2009 [28] | 37.92±8.26 | 36.46±7.53 | 25/17 | 24/17 | C14 | R3 | R1 | 10d | 10d |
| Cheng XH 2014 [29] | 45 | 45 | 75/45 | 75/45 | C13 | P1 | P3 | 7d | 7d |
| Deng YC 2017 [30] | 21~68 | 22~68 | 29/14 | 28/15 | C13 | E1 | O1 | 14d | 14d |
| Dong B 2015 [31] | 47.76±5.45 | 46.89±5.79 | 27/23 | 28/22 | C14 | E1 | O1 | 14d | 14d |
| Dong W 2013 [32] | NR | NR | NR | NR | RUT | O3 | O1 | 7d | 7d |
| Feng ML 2019 [33] | 45.21±7.36 | 45.08±7.23 | 34/39 | 35/38 | C14 | R1 | O1 | 14d | 14d |
| He MC 2022 [34] | 50.38±12.19 | 49.27±12.07 | 27/35 | 25/38 | C14 | R2 | R1 | 14d | 14d |
| He XJ 2021 [35] | 44.21±4.16 | 44.13±4.21 | 28/13 | 12/29 | NR | O1 | R1 | 14d | 14d |
| Hong LL 2017 [36] | 41.22±6.82 | 40.34±7.09 | 58/43 | 52/44 | C14 | R8 | R3 | 10d | 10d |
| Huang ZL 2016 [37] | 41.5±11.8 | 42.7±10.4 | 37/17 | 27/27 | C14 | R1 | P1 | 10d | 10d |
| Huang ZL 2016 [37] | 43.1±10.1 | | 23/32 | | C14 | O1 | | 10d | |
| Ji HB 2018 [38] | NR | NR | NR | NR | C13 | E2 | E1 | 10d | 10d |
| Kong CC 2014 [39] | 44.37±13.93 | 44.94±13.75 | 39/26 | 40/25 | C13 or C14 | E1 | E3 | 10d | 10d |
| Kong CC 2014 [39] | 45.88±13.74 | 43.69±13.90 | 33/32 | 34/31 | C13 and C14 | E2 | E4 | 10d | 10d |
| Kong SY 2020 [40] | 47.1±11.2 | 47.2±12.1 | 68/63 | 26/25 | C13 | E2 | E3 | 14d | 14d |
| Li CJ 2019 [41] | NR | NR | NR | NR | C14 | E1 | E2 | 14d | 14d |
| Li S 2015 [42] | 48.39±5.34 | 47.22±4.07 | 38/26 | 36/29 | C13 or C14 | R8 | R3 | 10d | 10d |
| Li S 2019 [43] | 44.93± 16.44 | 45.55± 15.00 | 32/28 | 33/27 | C13 or C14 | O9 | O2 | 14d | 14d |
| Li XT 2015 [44] | 44.2±3.6 | 44.4±2.2 | 32/18 | 32/18 | C13 | E1 | O1 | 10d | 10d |
| Li XY 2016 [45] | 46.3±4.2 | 49.7±3.8 | 57/52 | 58/51 | C13 | R3 | R1 | 14d | 14d |
| Liang C 2020 [46] | 46.30±6.57 | 45.77±6.91 | 29/19 | 28/20 | C14 | O1 | O3 | 14d | 14d |
| Liu SZ 2012 [47] | 49.2±16.5 | 49.2±16.5 | NR | NR | C14 | O1 | O3 | 7d | 7d |
| Liu SZ 2012 [47] | 49.2±16.5 | 49.2±16.5 | NR | NR | C14 | O2 | O8 | 7d | 7d |
| Liu YQ 2017 [48] | 18~70 | 18~70 | NR | NR | C13 | O1 | L1 | 14d | 14d |
| Liu YQ 2017 [48] | 18~70 | 18~70 | NR | NR | C13 | E1 | R1 | 14d | 14d |
| Lv X 2018 [49] | 50.12±10.69 | 46.35±13.05 | 20/20 | 16/15 | C13 | E1 | R1 | 14d | 14d |
| Ma HG 2016 [50] | 41.1±4.8 | 40.3±5.1 | 39/21 | 38/22 | ME | P1 | P3 | 10d | 10d |
| Qiao YG 2015 [51] | 18~60 | 18~60 | NR | NR | C13 | R2 | R8 | 7d | 7d |
| Ren J 2020 [52] | 52.43±2.02 | 52.03±2.77 | 24/17 | 23/18 | ME or C14 | O1 | E1 | 14d | 14d |
| Shen XH 2017 [53] | 19~73,46.8 | 23~73,47.4 | 20/25 | 20/25 | RUT and C13 | E2 | E1 | 10d | 10d |
| Shi J 2018 [54] | 38.45±9.04 | 39.17±8.75 | 31/21 | 28/19 | RUT | R1 | O1 | 14d | 14d |
| Wang JS 2013 [55] | 49.6±18.0 | 41.4±14.5 | 12/18 | 14/16 | C13 | R2 | R8 | 10d | 10d |
| Wang JX 2016 [56] | NR | NR | NR | NR | C14 | R8 | R3 | 14d | 14d |
| Wang WH 2017 [57] | 40.21±7.13 | 41.52±7.09 | 10/22 | 9/23 | ME | E2 | E1 | 10d | 10d |
| Wu X 2020 [58] | 45.24±10.26 | 44.96±11.27 | 26/25 | 30/24 | C13 or C14 | R1 | E1 | 10d | 10d |
| Xiong Y 2022 [59] | 45.17±5.45 | 46.13±5.62 | 14/18 | 17/15 | RUT and C14 | R1 | O1 | 14d | 14d |
| Xiong Y 2022 [59] | 45.66±5.71 | | 13/19 | | RUT and C14 | P1 | | 14d | |

*(Continued)*

**Table 1.** (Continued)

| Study ID | Age (Group 1) | Age (Group 2) | Female/Male (Group 1) | Female/Male (Group 2) | Method of Diagnosis | Intervention (Group 1) | Intervention (Group 2) | Duration (Group 1) | Duration (Group 2) |
|---|---|---|---|---|---|---|---|---|---|
| **Yang Q 2017 [60]** | 38.32±8.41 | 37.56±6.45 | 20/20 | 19/21 | C14 | E2 | E1 | 14d | 14d |
| **Yang QZ 2015 [61]** | 45.47 | 45.47 | NR | NR | C13 or C14 | L1 | L3 | 10d | 10d |
| **Yang WD 2021 [62]** | 68.4±1.2 | 68.5±1.3 | 32/23 | 30/25 | NR | O1 | E1 | 14d | 14d |
| **Yang X 2015 [63]** | 42.91±9.95 | 43.29±9.82 | 29/27 | 30/28 | C14 | R8 | R3 | 14d | 14d |
| **Yang X 2015 [63]** | 40.62±10.36 | | 28/27 | | C14 | R2 | | 14d | |
| **Yao YR 2011 [64]** | 40.11±12.56 | 37.65±11.23 | 16/13 | 13/11 | C14 | E2 | O1 | 7d | 7d |
| **Zeng H 2020 [65]** | 36.69±5.37 | 36.92±5.49 | 27/16 | 25/18 | RUT and C14 | L1 | E1 | 14d | 14d |
| **Zhai HZ 2016 [66]** | 41±13 | 39±11 | 76/54 | 68/52 | C13 | P1 | P3 | 7d | 7d |
| **Zhan SB 2020 [67]** | 47.52±3.56 | 48.54±3.69 | 27/23 | 28/22 | ME and C13 | R1 | E1 | 14d | 14d |
| **Zhan SB 2020 [67]** | 24~71 | 24~73 | 27/23 | 28/22 | ME and C13 | R1 | E1 | 14d | 14d |
| **Zhang HB 2013 [68]** | NR | NR | NR | NR | C14 | E3 | E1 | 7d | 7d |
| **Zhang HW 2013 [69]** | 41.3±10.6 | | 52/37 | | RUT and C14 | R2 | O1 | 10d | 10d |
| **Zhang MX 2014 [70]** | 47.90±13.49 | 47.98±10.98 | 40/40 | 44/36 | C13 or C14 | E1 | E3 | 14d | 14d |
| **Zhang MX 2014 [70]** | 47.37±11.92 | 48.84±11.61 | 38/37 | 39/40 | C13 or C14 | E6 | E2 | 14d | 14d |
| **Zhang WQ 2017 [71]** | 52.67±8.13 | 47.04±10.18 | 26/23 | 26/25 | C14 | L2 | L1 | 14d | 14d |
| **Zhang WQ 2017 [71]** | 45.59±9.07 | | 30/20 | | C14 | L3 | | 14d | |
| **Zhang XH 2021 [72]** | 48.89±12.24 | 47.26±14.05 | 70/36 | 68/38 | C13 | R1 | E1 | 14d | 14d |
| **Zhang XX 2022 [73]** | 52.09±13.32 | 50.85±12.59 | 39/40 | 31/48 | C13 or C14 | E2 | E1 | 14d | 14d |
| **Zhang Y 2018 [74]** | 52.14±10.19 | 52.09±10.94 | 38/22 | 39/21 | C14 | R1 | R2 | 10d | 10d |
| **Zhang Y 2018 [74]** | 52.71±12.36 | 52.59±11.32 | 41/19 | 37/23 | C14 | R4 | R7 | 10d | 10d |
| **Zhang YF 2022 [75]** | 42.91±13.05 | 43.07±11.95 | NR | NR | RUT | O1 | E1 | 14d | 14d |
| **Zheng ZZ 2021 [76]** | 32.58±3.18 | 32.41±3.24 | 35/33 | 34/34 | C14 | O1 | R1 | 14d | 14d |
| **Zhou X 2021 [77]** | 43.16±2.79 | 43.19±2.84 | 31/29 | 32/28 | C13 | R3 | O3 | 14d | 14d |
| **Zou J 2014 [78]** | 35.7±6.7 | 35.4±5.5 | 42/25 | 38/29 | C14 | E1 | O1 | 10d | 10d |

Table 1 summarizes key details of the included RCTs, including study ID, publication year, sample size, intervention and control measures. This provides an overview of the study design for each trial.

[a]Notes: NR: No Information; C13: C13 breath test; C14: C14 breath test; RUT: Rapid Urease Test; ME: Microscopic Examination; B: Bismuth; SUCRA: Surface under the cumulative ranking); RUT: Rapid Urease Test; CMT: Culture and Molecular Testing; AML: Amoxycillin; CLR: Clarithromycin; FRZ: Furazolidone; LEV: Levofloxacin; MTZ: Metronidazole; TE: Tetracycline; E1: Esomeprazole+AML+CLR; E2: Esomeprazole+AML+FRZ; E3: Esomeprazole+AML+LEV; L1:Lansoprazole+AML+CLR; L3:Lansoprazole+AML+LEV; O1: Omeprazole+AML+CLR; O3: Omeprazole+AML+LEV; P1: Pantoprazole+AML+CLR; P3: Pantoprazole+AML+LEV; R1: Rabeprazole+AML+CLR; R2: Rabeprazole+AML+FRZ; R3: Rabeprazole+AML+LEV; R8: Rabeprazole+FRZ+TE.

interventions was shown in Fig 4 (B). The league table of network meta-analysis was in Fig 6. Among the 55 studies included in the network meta-analysis, a total of 43 studies described adverse events, 6 studies described no adverse events, and 37 studies reported adverse events, specifically nausea, belching, fatigue, dizziness, bloating, diarrhea and a bitter taste in the mouth.

## Network meta-analyses of subgroup

In terms of eradication rates, the best interventions in the 7 regions were: Northeast-O1-75.98, North China-P3-61.75, East China-R8-76.85, South China-E2-68.17, Central China-R3-72.68, Northwest-E2-68.17, Southwest-R8-79.07. The worst interventions were: Northeast-E1-19.15, North China-O1-31.96, East China-O3-11.83, South China-L1-32.6, Central China-O1-10.81,

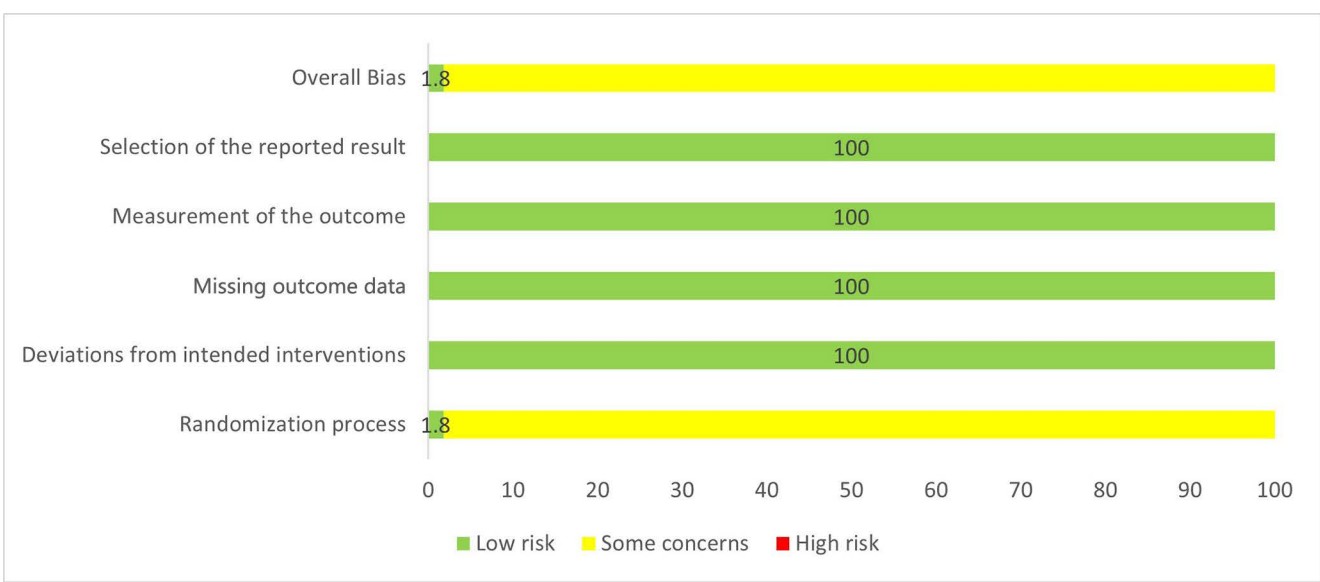

**Fig 2. The risk of bias graph.** This figure presents the proportion of included RCTs that were assessed as having low, unclear, or high risk of bias across six domains. The graph provides an overall visualization of the methodological quality of the included studies.

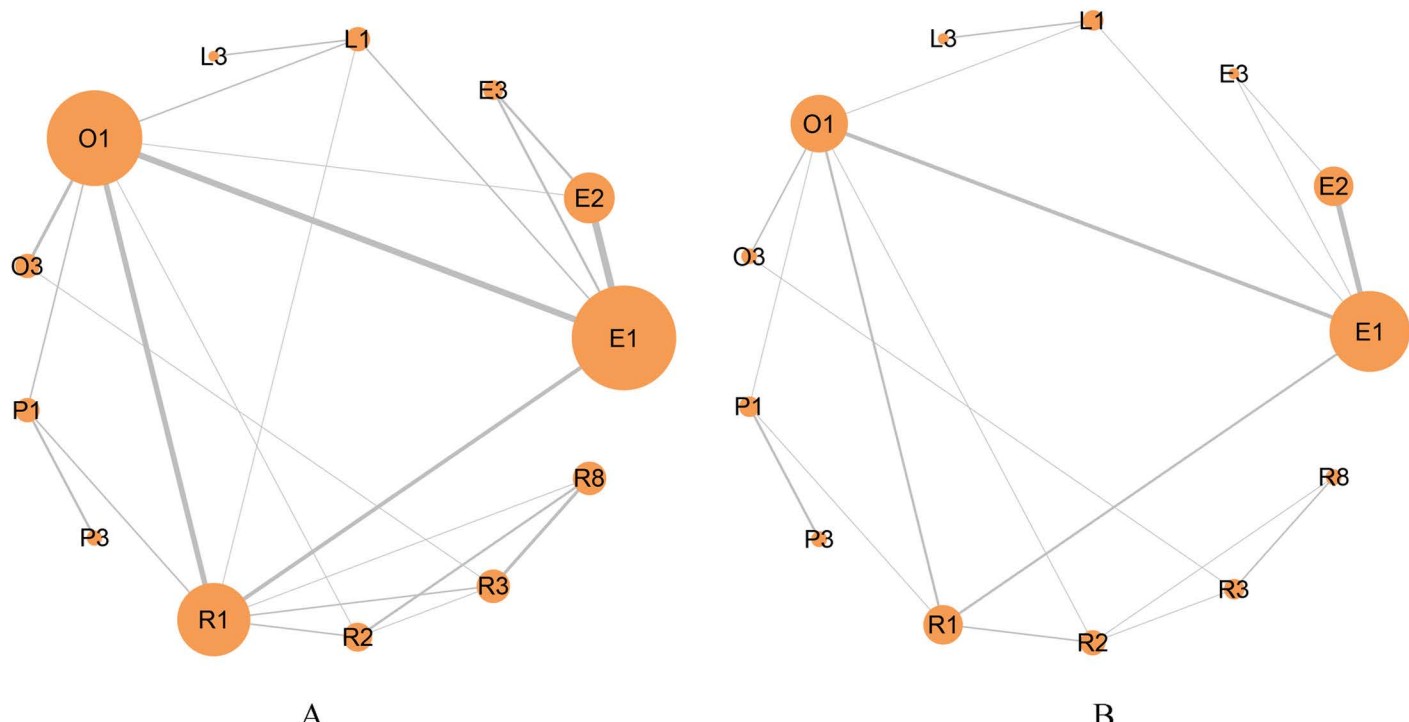

**Fig 3. The network comparison of interventions for primary and safety Outcomes.** The network diagram illustrates the comparative relationships among different intervention groups. The size of each circle represents the sample size of the corresponding intervention group, while the connecting lines indicate direct comparisons between groups. The thickness of the lines reflects the number of studies conducted for each comparison. Fig 3.A displays the network comparison for the primary outcome, and Fig 3.B represents the network comparison for the safety outcome.

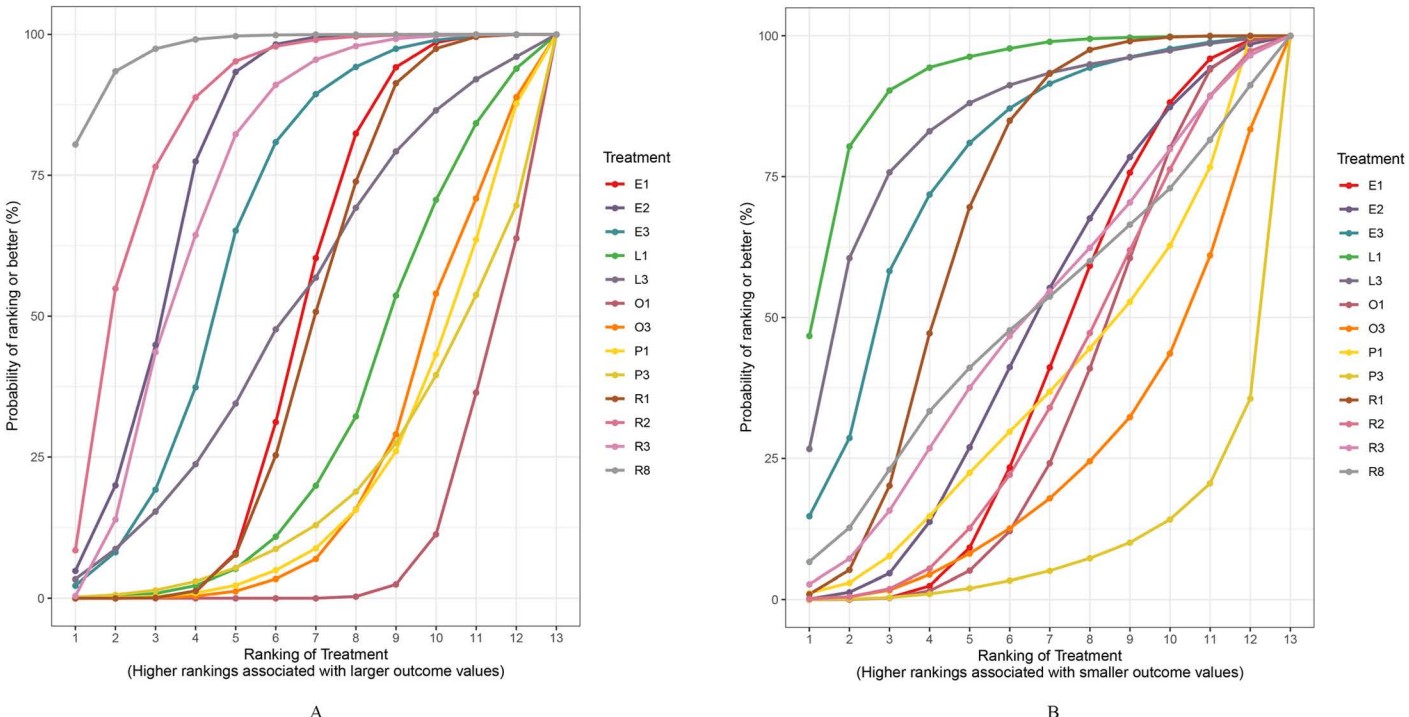

**Fig 4. The ranking probabilities of quadruple therapies for efficacy and safety.** The figure illustrates the ranking probabilities of different quadruple therapies. Each colored curve represents a specific therapy, with the area under the curve indicating its likelihood of being the most effective or safest. A larger area suggests better efficacy or safety. Fig 4.A presents the results for the primary outcome, while Fig 4.B shows the results for the safety outcome.

Northwest-L1-32.6, Southwest-O1-21.98. The network comparison map, ranking of interventions and the league table of each region can be found in S1 File.

## Discussion

In the network meta-analysis, we found that (i) The quadruple bismuth therapies for H. pylori recommended by the guidelines have a good performance in terms of eradication rate and safety; (ii) R8 (Rabeprazole + B + FRZ+TE) performed best on the outcome of H. pylori eradication rate and L1 (Lansoprazole + B + AML+CLR) on safety; (iii) Optimal interventions in eradication rate varied across regions.

For many infectious diseases, a single antibiotic is sufficient for treatment, but for persistent or slow-growing infections such as H. pylori, several concurrent and long-term antibiotics are generally required to kill persistent organisms and prevent the emergence of resistant microbial subpopulations. Once resistance has developed, the suitability of the preferred regimen is reduced, leading to the emergence of new treatment options. Due to the increasing resistance to antibiotics, the success of triple therapy consisting of PPI and two antibiotics has sharply decreased [79]. It is found that, when the H. pylori strain is tolerable and supersensitive, the rate of eradication of triple therapy is much lower [80]. In China, most physicians choose bismuth quadruple therapy (PPI+Bismuth+two antibiotics) as the initial treatment for H. pylori eradication nowadays [81,82].

Our study found that, among the treatments for H. pylori eradication, the R8 regimen (Rabeprazole + B + FRZ + TE) performed best in terms of eradication rate, while the L1 regimen (Lansoprazole + B + AML + CLR) showed the best safety profile. Rabeprazole has

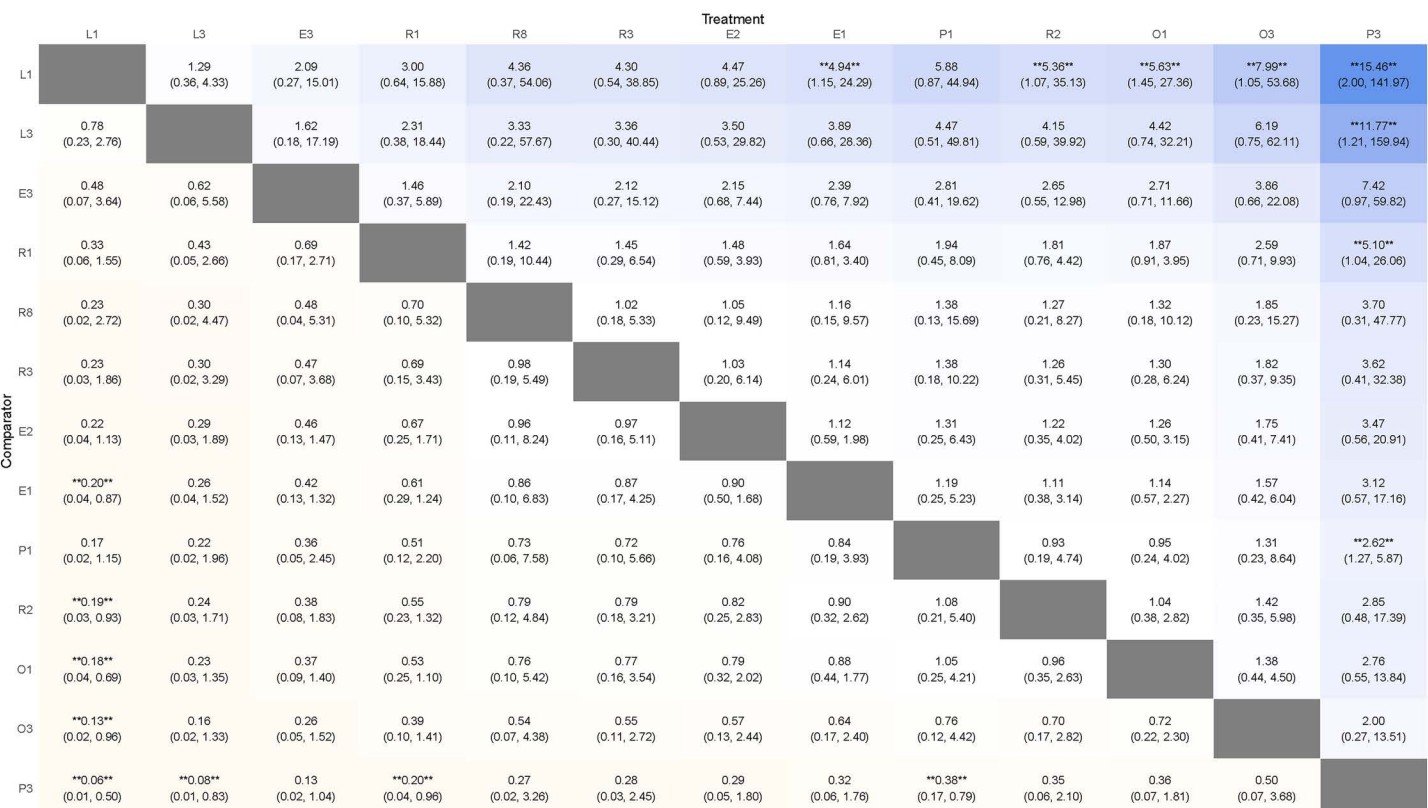

**Fig 5. The league table of network meta-analysis in primary outcome.** Each cell displays the results of comparisons between the "column intervention" and the "row intervention," expressed as [OR, 95% CI]. **** indicates statistical significance (P < 0.05). An OR > 1 suggests that the column intervention is more effective than the row intervention.

shown promising efficacy in the treatment of H. pylori infection [58,59,69,72]. The mechanism is related to its ability to inhibit gastric acid secretion and its antimicrobial properties [77,83]. PPIs have an inhibitory effect on gastric acid secretion and are currently used as first-line therapy for acid-related diseases [14,15,84]. Lansoprazole is a proton pump inhibitor with significant acid-suppressing effects, which raises the gastric pH and creates a more favorable environment for antibiotics to exert their bactericidal action. In the treatment of H. pylori infection, lansoprazole is commonly used in combination with antibiotics such as amoxicillin and clarithromycin to enhance H. pylori eradication rates [26]. Additionally, lansoprazole reduces gastric mucosal inflammation by inhibiting the production of inflammatory factors like TNF-α and IL-1β, which may further enhance the therapeutic efficacy against H. pylori infection [85]. This mechanism could explain why this study found that the combination of Lansoprazole and antibiotics exhibited the best safety profile.

Because of the limitations of new drugs, clinicians tend to get better efficacy by combining the original antibiotics differently [80]. A nationwide cross-sectional study in China has found the combination of AML+CLR and AML+LEV are two of the most often prescribed combinations by physicians for the eradication of H. pylori due to local resistance to antibiotics, guidelines, adverse response, and drug availability [81]. Otherwise, it is found that bismuth quadruple therapy combined with AML+TE has a high rate of eradication of H. pylori with 83% [86]. The resistance to antibiotics is an important factor for the eradication efficacy and

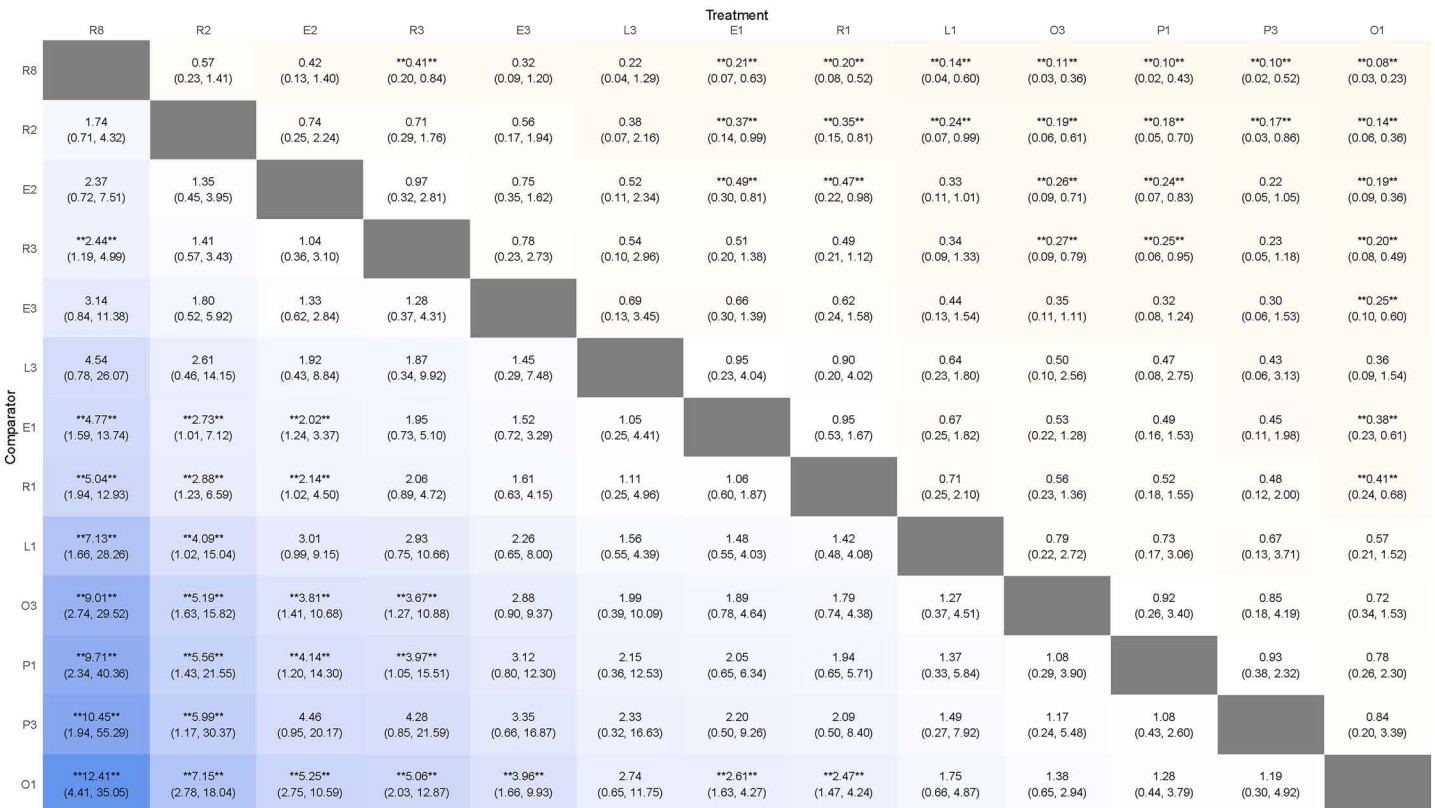

**Fig 6. The league table of network meta-analysis in safety outcome.** Each cell displays the results of comparisons between the "column intervention" and the "row intervention," expressed as [OR, 95% CI]. **** indicates statistical significance (P < 0.05). An OR < 1 suggests that the column intervention is more safe than the row intervention.

depends primarily on the geographical area [87,88]. For example, the eradication rate of H. pylori was higher significantly in Shanghai, China, than in Xi'an, China, with low resistance of CLR in Shanghai and high resistance in Xi'an [89]. Although the use of AML is long-term and worldwide, the resistance of AML remains low (1%-3%) [88,90]. In China, the resistance of CLR, with an increase from 15% in 2000 to 53% in 2014 [88] has greatly affected the treatment of H. pylori, which is one of the most important considerations [91]. As for TE, the probability of resistance is low(0.7%–1%) in most countries and for LEV in China it accounts for 6.7% [88]. Because of the vast territory of China, the resistance of antibiotics changes a lot. Subgroup analysis of geographical areas has found regional differences between the success of eradication in patients. Thus, the treatment schemes should be tailored according to the geographical characteristics. The results in our view also show different ranks in the outcomes of the eradication and safety of therapies in different areas of China.

Compared with the network meta-analysis [92,93] previously published, the advantages of this study were: first, we used the PPI and antibiotic types recommended in the Fifth consensus of China as the inclusion criteria for intervention. Although the guideline proposes a combination of 7 interventions, it does not limit the PPI in the combination. We included a total of 13 interventions for quadruple bismuth therapy according to the drug combinations in the included studies. Second, considering the different levels of antibiotic resistance in different regions of China, a subgroup analysis of seven major geographical regions in China was conducted. Moreover, in this study, we included 55 studies with a total of 11,682 patients,

and the efficacy ranking obtained through network meta-analysis was more convincing. Our limitations were that the included studies had a small sample size, which made the results obtained by each study less credible. The methodological qualities of the included studies were not high, with most of the studies not describing the randomization method in detail and not conducting blind studies with patients/doctors/statisticians. The low methodological quality may lead to bias in statistical results, resulting in reduced confidence in the results of the network meta-analysis.

In this study, the efficacy rankings of bismuth quadruple therapy in eradication rate and safety outcome were obtained through network meta-analysis, which can be referred to clinical practice. However, due to the small sample size and small number of studies included, it is still necessary to look forward to further validation of RCTs with larger sample sizes and higher methodological quality. Large-scale, multicenter studies with extended follow-up periods and comprehensive outcome measures are urgently needed to address current limitations in research and to elucidate the mechanisms underlying effective H. pylori treatment. Regional variations in infection prevalence and antibiotic resistance highlight the necessity of geographically tailored analyses to optimize therapeutic outcomes. Moreover, this study found that among the 130 treatment arms included in the network analysis, 27 arms demonstrated an eradication rate of less than 80%. These arms predominantly involved combinations of different PPIs with clarithromycin or levofloxacin, reflecting the high resistance rates to clarithromycin and levofloxacin reported in mainland China [4]. This highlights the inadequacy of current methods for H. pylori eradication. The 2022 Chinese National Clinical Practice Guideline [94] on Helicobacter pylori Eradication Treatment recommends both PPI-based and P-CAB-based bismuth quadruple therapies as first-line and salvage treatment options for H. pylori infection. Similarly, the Fifth Consensus Report in China [4] notes that the novel potassium-competitive acid blocker (P-CAB) vonoprazan provides stronger acid suppression compared to existing PPIs, which could potentially enhance H. pylori eradication rates. Hence, incorporating the new generation of acid-suppressing drugs into bismuth quadruple regimens offers a promising avenue for improving treatment efficacy, warranting further investigation through rigorous trials. Individualized treatment strategies, informed by local resistance patterns and patient-specific factors, should be emphasized alongside long-term monitoring to ensure sustained eradication and minimize recurrence. These combined efforts will enhance eradication rates, reduce antibiotic resistance, and provide robust evidence to refine clinical guidelines.

## Conclusions

In summary, we found that the quadruple bismuth therapy recommended by the guidelines performed well on eradication rates and safety outcome by network meta-analysis. Combining the results of subgroup analyses in different regions of China shows that the best interventions for eradication rates are not the same due to differences in antibiotic resistance between regions. Because of the important issue of antibiotics use, it is essential to establish a local databank on the efficacy of regimens to assist clinicians in selecting the optimal treatment. At the same time, multi-regional, multi-center, and large-sample-size RCTs need to be performed to obtain more direct clinical evidence, particularly focusing on new H. pylori eradication therapies incorporating the new generation of acid-suppressing drugs into bismuth quadruple regimens.

## Supporting information

**S1 Table. Search strategies.** The table of search strategies.
(DOCX)

**S2 Table. Inclusion and Exclusion literatures.** The characteristics of all inclusion and exclusion literatures.
(XLSX)

**S3 Table. The table of risk of bias assessment.** The results of the risk of bias assessment for each domain of the included trials.
(XLSX)

**S1 File. The network comparison map, ranking of interventions and the league table of each region.** The network comparison map, ranking of interventions and the league table of 7 regions (including Northeast, North China, East China, South China, Central China, Northwest and Southwest).
(RAR)

## Author contributions

**Conceptualization:** Jiali Wei, Zehao Zheng, Boyi JIA, Jiayi WANG, Mei Han.

**Data curation:** Jiali Wei, Zehao Zheng, Xin WANG, Boyi JIA, Mingyao Sun, Jiayi WANG, Qin WAN, Mei Han.

**Formal analysis:** Zehao Zheng, Boyi JIA, Mei Han.

**Funding acquisition:** Mei Han.

**Investigation:** Jiali Wei, Yue Qiu.

**Methodology:** Jiali Wei, Xin WANG, Yue Qiu.

**Project administration:** Jiali Wei, Mei Han.

**Software:** Jiali Wei, Zehao Zheng, Mingyao Sun, Qin WAN.

**Supervision:** Mei Han, Yue Qiu.

**Validation:** Xin WANG, Mingyao Sun, Jiayi WANG, Qin WAN.

**Visualization:** Zehao Zheng, Xin WANG, Boyi JIA, Mingyao Sun.

**Writing – original draft:** Jiali Wei, Zehao Zheng.

**Writing – review & editing:** Jiali Wei, Zehao Zheng, Mei Han, Yue Qiu.

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
