## [Decision Letter · Decision Letter 0]

18 Nov 2024

PONE-D-24-31187Guideline-based for Helicobacter pylori infection in China: A Systematic Review and Network Meta-analysis of Bismuth Quadruple TherapyPLOS ONE

Dear Dr. Han,

Thank you for submitting your manuscript to PLOS ONE. After careful consideration, we feel that it has merit but does not fully meet PLOS ONE’s publication criteria as it currently stands. Therefore, we invite you to submit a revised version of the manuscript that addresses the points raised during the review process.

We look forward to receiving your revised manuscript.

Kind regards,

Addisu Melese Dagnaw, MSc

Academic Editor

PLOS ONE

Journal requirements:    When submitting your revision, we need you to address these additional requirements. 1. Please ensure that your manuscript meets PLOS ONE's style requirements, including those for file naming. The PLOS ONE style templates can be found at https://journals.plos.org/plosone/s/file?id=wjVg/PLOSOne_formatting_sample_main_body.pdf and https://journals.plos.org/plosone/s/file?id=ba62/PLOSOne_formatting_sample_title_authors_affiliations.pdf 2. As required by our policy on Data Availability, please ensure your manuscript or supplementary information includes the following:  A numbered table of all studies identified in the literature search, including those that were excluded from the analyses.   For every excluded study, the table should list the reason(s) for exclusion.   If any of the included studies are unpublished, include a link (URL) to the primary source or detailed information about how the content can be accessed.  A table of all data extracted from the primary research sources for the systematic review and/or meta-analysis. The table must include the following information for each study:  Name of data extractors and date of data extraction  Confirmation that the study was eligible to be included in the review.   All data extracted from each study for the reported systematic review and/or meta-analysis that would be needed to replicate your analyses.  If data or supporting information were obtained from another source (e.g. correspondence with the author of the original research article), please provide the source of data and dates on which the data/information were obtained by your research group.  If applicable for your analysis, a table showing the completed risk of bias and quality/certainty assessments for each study or outcome.  Please ensure this is provided for each domain or parameter assessed. For example, if you used the Cochrane risk-of-bias tool for randomized trials, provide answers to each of the signalling questions for each study. If you used GRADE to assess certainty of evidence, provide judgements about each of the quality of evidence factor. This should be provided for each outcome.   An explanation of how missing data were handled.   This information can be included in the main text, supplementary information, or relevant data repository. Please note that providing these underlying data is a requirement for publication in this journal, and if these data are not provided your manuscript might be rejected.   3. PLOS requires an ORCID iD for the corresponding author in Editorial Manager on papers submitted after December 6th, 2016. Please ensure that you have an ORCID iD and that it is validated in Editorial Manager. To do this, go to ‘Update my Information’ (in the upper left-hand corner of the main menu), and click on the Fetch/Validate link next to the ORCID field. This will take you to the ORCID site and allow you to create a new iD or authenticate a pre-existing iD in Editorial Manager. 4. Thank you for stating the following financial disclosure:  [High-level traditional Chinese medicine key subjects construction project of National Administration of Traditional Chinese Medicine——Evidence-based Traditional Chinese Medicine (zyyzdxk-2023249) and the Basic Research funding of Beijing University of Chinese Medicine.].  Please state what role the funders took in the study.  If the funders had no role, please state: ""The funders had no role in study design, data collection and analysis, decision to publish, or preparation of the manuscript."" If this statement is not correct you must amend it as needed. Please include this amended Role of Funder statement in your cover letter; we will change the online submission form on your behalf. 5. Please include captions for your Supporting Information files at the end of your manuscript, and update any in-text citations to match accordingly. Please see our Supporting Information guidelines for more information: http://journals.plos.org/plosone/s/supporting-information.  6. Please include a copy of Table 1 which you refer to in your text on page 10.

Reviewers' comments:

Reviewer's Responses to Questions

**Comments to the Author**

1. Is the manuscript technically sound, and do the data support the conclusions?

Reviewer #1: Yes

Reviewer #2: Yes

Reviewer #3: Partly

2. Has the statistical analysis been performed appropriately and rigorously? 

Reviewer #1: Yes

Reviewer #2: Yes

Reviewer #3: No

3. Have the authors made all data underlying the findings in their manuscript fully available?

Reviewer #1: Yes

Reviewer #2: Yes

Reviewer #3: No

4. Is the manuscript presented in an intelligible fashion and written in standard English?

Reviewer #1: Yes

Reviewer #2: No

Reviewer #3: Yes

5. Review Comments to the Author

Reviewer #1: Bismuth quadruple treatments for H. pylori infection are recommended by The Sixth Consensus Report in China. This manuscript is interesting.

1.It is better to add studies prescribing vonoprazan based bismuth-containing quadruple therapy which is recommended by The Sixth Consensus Report in China in the manuscript.

2.Some studies reported with unsatisfactory eradication rates（＜80%）should be included in the "DISCUSSION" section.

Reviewer #2: In this study, the authors analyzed the priority for efficacy and safety of bismuth quadruple regimens on management of Helicobacter pylori infection in China. The study found that the antibiotic combination of tetracycline and furazolidone achieved the best efficacy and the antibiotic combination of amoxicillin and clarithromycin achieved the best safety. To further enhance the quality of the article, I have the following comments:

Comments:

1. Reference 4, the “Sixth Consensus” cited in the article was stated as “non-eradication treatment part”, while quadruple therapy regimens referred to in the article were in the “treatment part”1 of the consensus. Please clarify whether the literature has been misquoted. Actually, it should be called a guideline.

1Zhou L, Lu H, Song Z, et al. 2022 Chinese national clinical practice guideline on Helicobacter pylori eradication treatment [published correction appears in Chin Med J (Engl). 2024 May 5;137(9):1068. doi: 10.1097/CM9.0000000000003134]. Chin Med J (Engl). 2022;135(24):2899-2910. Published 2022 Dec 20. doi:10.1097/CM9.0000000000002546

2. The inclusion criterion “regardless of age” described in the article seems to contradict the exclusion criterion “for children” in Figure 1.

3. Heterogeneity is only mentioned in “Analysis Methods”, so how heterogeneous was the data in outcome analysis?

4.The emergence of potassium-competitive acid blockers such as vonoprazan has brought new life to H. pylori treatment, and the impact of the new generation of acid-suppressing drugs on bismuth quadruple regimens should be discussed.

5.Please recheck Figure 1 for the accuracy of the number of articles, and provide evidence of literature screening if necessary to ensure the quality of the literature search.

6.The background is too lengthy. Focus on the 2022 guidelines' recommendations for bismuth quadruple therapy and simplify accordingly.

7.The article should be proofread by a native English speaker, especially to avoid mistakes like “DISSCUSSION.”

8.The author should cite the latest pooled resistance data from China, emphasizing region-specific strategies, and refer to the updated principles of the 2024 ACG clinical guidelines.

9.Figures should be modified and formatted as necessary to ensure logical flow and an aesthetically pleasing presentation.

10.It is necessary to present appropriate tables and forest plots to display effect sizes and rank specific therapies. For example, the best strategies by region. Please reconsider the presentation format.

11.Why emphasize the four-week eradication rate in the conclusion? Did the authors ensure that the included studies all report the eradication rate after only four weeks?

12.Can subgroup analyses be conducted for first-line and rescue therapies?

13.The discussion should focus on the research findings rather than repeating the background.

14.Based on the study results of this article, what evidence-based recommendations should be made for the future?

Reviewer #3: The results should present the effect sizes of H. pylori eradiation and adverse event rates instead of SUCRA in the manuscript, which make the main findings difficult to understand for the readers. Besides, the comparisons among different antibiotic regimens and different PPIs were recommended by subgroup analysis or component NMA. Further more the interaction between different PPIs and antibiotic regimens may need adjustment by meta-regression.

6. PLOS authors have the option to publish the peer review history of their article (what does this mean? ). If published, this will include your full peer review and any attached files.

**Do you want your identity to be public for this peer review?** For information about this choice, including consent withdrawal, please see our Privacy Policy .

Reviewer #1: No

Reviewer #2: No

Reviewer #3: **Yes: ** Tun-Chieh Chen

---

## [Author Response · Author response to Decision Letter 0]

5 Jan 2025

Title: Guideline-based for Helicobacter pylori infection in China: A Systematic Review and Network Meta-analysis of Bismuth Quadruple Therapy

Authors: Jiali Wei, Zehao Zheng, Xin Wang, Boyi Jia, Mingyao Sun, Jiayi Wang, Qin Wan, Mei Han, Yue Qiu

Manuscript lD: PONE-D-24-31187

Dear Editors,

Thank you for giving us the opportunity to submit a revised draft of the manuscript titled "Guideline-based for Helicobacter pylori infection in China: A Systematic Review and Network Meta-analysis of Bismuth Quadruple Therapy" for publication in PLOS ONE. We deeply appreciate the time and effort that you and the reviewers dedicated to providing feedback on our manuscript, and we are grateful for the insightful comments and valuable suggestions that have significantly improved our paper.

We have carefully considered all the feedback provided and have made revisions accordingly. The changes made to the manuscript are highlighted within the marked-up copy. We believe that these revisions have strengthened the clarity and quality of the manuscript, and we are confident that it is now in a much better form for publication.

As requested, we are submitting the following materials:

1. A rebuttal letter responding to each point raised by the academic editor and reviewer(s).

2. A marked-up copy of the manuscript, with all changes highlighted.

3. An unmarked version of the revised manuscript, with no tracked changes.

Below, we have outlined the specific modifications made in response to the reviewers' comments:

1. Our manuscript is now fully complies with PLOS ONE's style requirements.

2. The [Inclusion and exclusion literature Data Sheet] is now included in the supplementary materials, as Supplementary Table S2.

3. The ORCID ID for the corresponding author is provided.

4. The following statement has been added to the manuscript: "The funders had no role in study design, data collection and analysis, decision to publish, or preparation of the manuscript."

5. Captions for the Supporting Information files have been included at the end of the manuscript.

6. A copy of Table 1 has been included on page 10.

Thank you for your careful review and for the valuable suggestions provided by Reviewer 1. In response to the reviewer's comments, I have made the following revisions and clarifications:

1. Regarding the mention of "the new potassium-competitive acid blocker Vonoprazan," as indicated in The Fifth Consensus Report in China, which states that Vonoprazan has a stronger acid-suppressive effect compared to existing PPIs and may further improve the H. pylori eradication rate: According to the 2022 "Chinese National Clinical Practice Guideline on Helicobacter pylori Eradication Treatment," both the bismuth-containing quadruple therapy with PPI and the bismuth-containing quadruple therapy with P-CAB are recommended as first-line and second-line treatments for H. pylori eradication (weak recommendation, low-quality evidence). However, in the network meta-analysis conducted in this study, we selected PPIs based on higher-quality evidence from the guideline, including esomeprazole, rabeprazole, omeprazole, lansoprazole, pantoprazole, and esomeprazole. Moreover, we based our analysis on the most commonly used combinations in clinical research. We did not retrieve any randomized controlled trials conducted in mainland China on vonoprazan, and we were unable to include it in the network analysis. If we have overlooked any relevant studies, we would appreciate it if you could recommend additional references for inclusion.

2. Regarding the eradication rate of less than 80%, this has already been addressed in the discussion section. Among the 130 treatment arms included in the network analysis, 27 (20.8%) had an eradication rate of less than 80%. These arms were primarily combinations of different PPIs with clarithromycin or levofloxacin, consistent with the high resistance rates to these antibiotics reported in mainland China. This underscores the inadequacy of current H. pylori eradication methods.

Thank you for your thorough review and for the valuable comments provided by Reviewer 2. In response to the reviewer’s suggestions, I have made the following revisions and clarifications:

1. Regarding the mention of "The Sixth Consensus Report in China," you pointed out that it primarily emphasizes non-eradication treatment methods and diagnostic content, while the relevant H. pylori therapies are discussed in detail in "The Fifth Consensus Report in China," and both consensus reports make similar recommendations for quadruple therapy. In light of this, we have updated the reference to "The Fifth Consensus Report in China."

2. Regarding the inclusion criterion “regardless of age” mentioned in the article, which seems to contradict the exclusion criterion “for children” in Figure 1, we reviewed the data extraction table and found that our study did not include children. Therefore, we have corrected the age-related inclusion criterion.

3. Concerning the distinction between heterogeneity and inconsistency, we have clarified the issue. Heterogeneity refers to differences between studies in direct comparisons, while inconsistency refers to differences in indirect comparisons within the network. Inconsistency can be viewed as a form of heterogeneity within the network structure. As the number of multi-arm studies increases, inconsistency tends to decrease. In our study, multi-arm studies accounted for over 27% (15/55), thus the inconsistency is relatively low. We reviewed published network meta-analyses and relevant textbooks, and based the description of heterogeneity (inconsistency) primarily on the P-values and I² values in the forest plots. Given that our network meta-analysis included 13 intervention measures, resulting in large forest plots, and no common control measures, this study used the league table of network meta-analysis to present the pairwise comparisons of any interventions using OR, 95% CI, and p-values.

4. Regarding the impact of new-generation acid-suppressing drugs on bismuth quadruple regimens, we have elaborated on this point in the discussion. "The Fifth Consensus Report in China" mentions that "Vonoprazan, a new potassium-competitive acid blocker, has a stronger acid-suppressive effect than existing PPIs, and its application is expected to further improve the H. pylori eradication rate." The 2022 "Chinese National Clinical Practice Guideline on Helicobacter pylori Eradication Treatment" states that both bismuth quadruple therapy containing PPI and bismuth quadruple therapy containing P-CAB are recommended for initial and second-line H. pylori eradication treatment (weak recommendation, low-quality evidence). Due to the limited RCTs on Vonoprazan in mainland China, it was not included in our study. We believe that future clinical studies on bismuth quadruple therapy with P-CAB and PPI should be conducted to further verify their clinical efficacy.

5. Figure 1 has been corrected, and relevant details regarding the inclusion and exclusion of studies in the full-text screening process have been provided.

6. The background section has been simplified: As per your recommendation, we have focused on the 2022 guidelines' recommendations for bismuth quadruple therapy and simplified the background accordingly.

7. The manuscript has been proofread: The paper has been reviewed by a native English speaker to ensure clarity and correctness of the language.

8. The latest Chinese antibiotic resistance data has been cited in the introduction: We have emphasized the necessity of performing subgroup analysis based on regional differences, and we also referred to the 2024 ACG Clinical Guidelines on Helicobacter pylori infection, noting the geographical and racial differences in Hp infection in the U.S. The revised text reads as follows:

"Due to varying levels of antibiotic resistance and environmental factors, regional differences can significantly impact the effectiveness of Hp eradication therapies. For instance, the prevalence and acquisition patterns of Hp infection often correlate with socioeconomic and demographic factors, such as population density, hygiene standards, and racial composition. These differences suggest that treatment outcomes may not be uniform across regions. In China, where the geographical diversity encompasses variations in healthcare access, dietary habits, and antibiotic usage, a subgroup analysis by region becomes essential. This approach allows for a more accurate evaluation of treatment regimens, ensuring tailored strategies that address the unique characteristics and challenges of each region."

9. Figures and tables have been revised and formatted: Necessary adjustments and formatting have been made to ensure the logical flow and visual appeal of the figures and tables.

10. Regarding the display of network meta-analysis results: Since the study included 13 interventions in the network meta-analysis, it was not possible to obtain a forest plot for the comparisons of these interventions using the Bayesian model. Therefore, we used the league table of network meta-analysis to present the comparisons between the "row" and "column" interventions (i.e., comparisons among all interventions). Previously, due to the large size of the league table, we had placed it in the supplementary materials; however, it has now been moved to the main manuscript, with an efficacy ranking chart added to display the effect sizes and rank specific therapies.

11. Regarding treatment duration and H. pylori eradication rate: According to The Fifth Consensus Report in China, the recommended treatment duration is 7-14 days, with C13 or C14 testing conducted 4 weeks after treatment to assess H. pylori levels. All studies included in our data extraction reported the 4-week eradication rate as the outcome indicator.

12. Regarding the rescue regimens: The Fifth Consensus Report in China recommends that the choice of rescue regimens should take previous first-line treatments into account, with the principle of avoiding the reuse of prior regimens. This systematic review and network meta-analysis compare the clinical efficacy and safety of the first-line H. pylori eradication regimen recommended by Chinese guidelines. All studies included in this review focus on first-line therapies, which also serve as rescue regimens.

13. Enhancements to the Discussion section: I have revised the Discussion section, focusing on summarizing and interpreting the study results more clearly, and further elaborating on the clinical significance of our findings.

14. Future clinical recommendations: Based on the findings of this study, we have outlined the following future clinical strategies: Future strategies for Helicobacter pylori (Hp) eradication should prioritize evidence-based approaches and clinical innovations. Large-scale, multicenter studies with extended follow-up periods and comprehensive outcome measures are urgently needed to address current limitations in research and to elucidate the mechanisms underlying effective Hp treatment. Regional variations in infection prevalence and antibiotic resistance highlight the necessity of geographically tailored analyses to optimize therapeutic outcomes. Clinically, incorporating the new generation of acid-suppressing drugs into bismuth quadruple regimens offers a promising avenue for improving treatment efficacy, warranting further investigation through rigorous trials. Individualized treatment strategies, informed by local resistance patterns and patient-specific factors, should be emphasized alongside long-term monitoring to ensure sustained eradication and minimize recurrence. These combined efforts will enhance eradication rates, reduce antibiotic resistance, and provide robust evidence to refine clinical guidelines.

Thank you for your thorough review and the valuable suggestions provided by Reviewer 3. In response to your feedback, I have made the following revisions:

1. Modification to the presentation of network meta-analysis results: Since this study included 13 interventions, the Bayesian model could not generate a forest plot for pairwise comparisons. Therefore, we decided to present the results using the league table of network meta-analysis to compare the “column” treatments with the “row” treatments. The results of the comparisons between interventions are now expressed using OR and 95% CI. Additionally, due to the large size of the the league table of network meta-analysis, it was initially placed in the supplementary material; it has now been moved to the main text and supplemented with an efficacy ranking chart and area under the curve (AUC) to display the effect sizes and rank the treatments accordingly.

2. Modification to intervention arm analysis and ranking: Each arm in the network meta-analysis represents a combination of PPI and two antibiotics, and efficacy rankings have been provided for them. For example: E1 (Esomeprazole + B + AML + CLR); E2 (Esomeprazole + B + AML + FRZ); E3 (Esomeprazole + B + AML + LEV); L1 (Lansoprazole + B + AML + CLR); L3 (Lansoprazole + B + AML + LEV); O1 (Omeprazole + B + AML + CLR); O3 (Omeprazole + B + AML + LEV); P1 (Pantoprazole + B + AML + CLR); P3 (Pantoprazole + B + AML + LEV); R1 (Rabeprazole + B + AML + CLR); R2 (Rabeprazole + B + AML + FRZ); R3 (Rabeprazole + B + AML + LEV); R8 (Rabeprazole + B + FRZ + TE). The study found that, among the treatments for H. pylori eradication, the R8 regimen (Rabeprazole + B + FRZ + TE) performed best in terms of eradication rate, while the L1 regimen (Lansoprazole + B + AML + CLR) showed the best safety profile.

Once again, we sincerely thank you and the reviewers for your time and valuable contributions to our manuscript. Should you have any further questions or require additional revisions, please do not hesitate to contact us.

Kind regards,

Mei Han

Centre for Evidence-Based Medicine, Beijing University of Chinese Medicine, Beijing, 100029, China.

---

## [Editor Report · Decision Letter 1]

21 Jan 2025

PONE-D-24-31187R1Guideline-based for Helicobacter pylori infection in China: A Systematic Review and Network Meta-analysis of Bismuth Quadruple TherapyPLOS ONE

Dear Dr. Han,

Thank you for submitting your manuscript to PLOS ONE. After careful consideration, we feel that it has merit but does not fully meet PLOS ONE’s publication criteria as it currently stands. Therefore, we invite you to submit a revised version of the manuscript that addresses the points raised during the review process.

The manuscript title "Guideline-based for Helicobacter pylori infection in China: A Systematic Review and Network Meta-analysis of Bismuth Quadruple Therapy" has few confusions and needs a kind of revision to give comprehensive understanding. I recommend modifying it as: "Guideline-based Bismuth Quadruple Therapy for Helicobacter pylori infection in China: A Systematic Review and Network Meta-analysis" or consult language experts otherwise.

We look forward to receiving your revised manuscript.

Kind regards,

Addisu Melese Dagnaw, MSc

Academic Editor

PLOS ONE

Journal Requirements:

Additional Editor Comments :

The manuscript title "Guideline-based for Helicobacter pylori infection in China: A Systematic Review and Network Meta-analysis of Bismuth Quadruple Therapy" has few confusions and needs a kind of revision to give comprehensive understanding. I recommend modifying it as: "Guideline-based Bismuth Quadruple Therapy for Helicobacter pylori infection in China: A Systematic Review and Network Meta-analysis" or consult language experts otherwise.
---

## [Author Response · Author response to Decision Letter 1]

21 Jan 2025

Response to Reviewers

Title: Guideline-based Bismuth Quadruple Therapy for Helicobacter pylori infection in China: A Systematic Review and Network Meta-analysis

Authors: Jiali Wei, Zehao Zheng, Xin Wang, Boyi Jia, Mingyao Sun, Jiayi Wang, Qin Wan, Mei Han, Yue Qiu

Manuscript lD: PONE-D-24-31187

Dear Editors,

Thank you for giving us the opportunity to submit a revised draft of the manuscript titled "Guideline-based Bismuth Quadruple Therapy for Helicobacter pylori infection in China: A Systematic Review and Network Meta-analysis" for publication in PLOS ONE. We deeply appreciate the time and effort that you and the reviewers dedicated to providing feedback on our manuscript, and we are grateful for the insightful comments and valuable suggestions that have significantly improved our paper.

We have carefully considered all the feedback provided and have made revisions accordingly. The changes made to the manuscript are highlighted within the marked-up copy. We believe that these revisions have strengthened the clarity and quality of the manuscript, and we are confident that it is now in a much better form for publication.

As requested, we are submitting the following materials:

1. A rebuttal letter responding to each point raised by the academic editor and reviewer(s).

2. A marked-up copy of the manuscript, with all changes highlighted.

3. An unmarked version of the revised manuscript, with no tracked changes.

Below, we have outlined the specific modifications made in response to the comments:

1. We have updated the title to: "Guideline-Based Bismuth Quadruple Therapy for Helicobacter pylori Infection in China: A Systematic Review and Network Meta-Analysis."

2. We have thoroughly reviewed the reference list to ensure its accuracy and completeness.

Once again, we sincerely thank you for your time and valuable contributions to our manuscript. Should you have any further questions or require additional revisions, please do not hesitate to contact us.

Kind regards,

Mei Han

Centre for Evidence-Based Medicine, Beijing University of Chinese Medicine, Beijing, 100029, China.

---

## [Editor Report · Decision Letter 2]

24 Jan 2025

Guideline-based Bismuth Quadruple Therapy for Helicobacter pylori infection in China: A Systematic Review and Network Meta-analysis

PONE-D-24-31187R2

Dear Dr. Han,

We’re pleased to inform you that your manuscript has been judged scientifically suitable for publication and will be formally accepted for publication once it meets all outstanding technical requirements.

Kind regards,

Addisu Melese Dagnaw, MSc

Academic Editor

PLOS ONE
---

## [Editor Report · Acceptance letter]

PONE-D-24-31187R2

PLOS ONE

Dear Dr. Han,

I'm pleased to inform you that your manuscript has been deemed suitable for publication in PLOS ONE. Congratulations! Your manuscript is now being handed over to our production team.

Kind regards,

on behalf of

Mr. Addisu Melese Dagnaw

Academic Editor

PLOS ONE